# Trends in State Anxiety during the Full Lockdown in Italy: The Role Played by COVID-19 Risk Perception and Trait Emotional Intelligence

Elisa Tedaldi [1,†], Noemi Orabona [1,2,†], Ani Hovnanyan [1], Enrico Rubaltelli [1] and Sara Scrimin [1,*]

[1]  Department of Developmental Psychology and Socialization, University of Padova, Via Venezia 8, 35131 Padova, Italy; elisa.tedaldi@phd.unipd.it (E.T.); noemi.orabona@unitn.it (N.O.); ani.hovnanyan@phd.unipd.it (A.H.); enrico.rubaltelli@unipd.it (E.R.)

[2]  Department of Psychology and Cognitive Science, University of Trento, Corso Bettini 81, 38068 Trento, Italy

[*]  Correspondence: sara.scrimin@unipd.it; Tel.: +39-04-9827-1275

[†]  These authors contributed equally to this work.

**Abstract:** The COVID-19 pandemic is associated with mental health outcomes in the general population. This study assessed how state anxiety changed at different time points during the pandemic and how it was influenced by risk perception and trait emotional intelligence (trait EI). The study was conducted online in two data collections, at the beginning (wave 1, N = 1031) and at the end (wave 2, N = 700) of the lockdown. Participants were asked to self-report their state anxiety, risk perception of COVID-19 contagiousness, and trait EI. The interaction between risk perception and wave showed that, in wave 1 (but not in wave 2), anxiety increased as risk perception increased. Further, trait EI by wave interactions showed that effective (vs. ineffective) regulators experienced lower anxiety and this difference was larger in wave 2 than in wave 1. Because of the cross-sectional design of the study and the convenience sample we should be cautious when generalizing the present findings to the entire population. Our findings support the moderating role of trait EI on state anxiety during the COVID-19 pandemic. This knowledge provides further support for the importance of EI in coping with uncertain and stressful environmental conditions such as those posed by the COVID-19 pandemic.

**Keywords:** COVID-19; mental health; anxiety; trait EI; risk perception

## 1. Introduction

The coronavirus outbreak in Italy started on 23 February 2020, reaching a huge number of positive cases and deaths in a short period of time. The government was, eventually, forced to impose a nationwide lockdown for two months [1]. Epidemics are known to trigger fear and anxiety among the population potentially causing changes in behaviors and psychological distress [2]. In Italy, since the early stages of the pandemic, people experienced a high degree of uncertainty associated with significant stress responses. In fact, uncertainty is regarded as a powerful trigger of stress response and worrisome thinking [3,4]. The literature on risk perception shows that these responses were likely induced by the novelty of the virus, the presence of asymptomatic patients, and the lack of scientific knowledge on how to fight it (e.g., a vaccine) [5]. Risk perception is defined as people's tendency to rely on intuitive risk judgments and to experience risk in vicarious ways, for instance through the news media and their thorough documentation of the hazards occurring around the world. These evaluations are often very different from those resulting from the sophisticated risk assessments employed by experts [3]. Since the pandemic was likely perceived as a hazard, it is not surprising that recent studies have documented an association between increased risk perception related to COVID-19 and heightened levels of anxiety as well as other negative mental health problems [6–9].

For instance, between 27 March and 6 April 2020, adults living in Italy reported high state anxiety (20.8%), post-traumatic stress symptoms (37%), depression (17.3%), insomnia (7.3%), high perceived stress (21.8%) and adjustment disorder symptoms (22.9%) [8].

Despite the increase in mental health problems as a response to the pandemic, not all individuals display psychological distress. Community trauma research has shown how people can be resilient and positively adjust when forced to deal with potentially traumatic events [10,11]. Several factors may play a role in fostering resilience and adaptation [12]. One of these is the individuals' ability to regulate emotions. Emotion regulation refers to individuals' capacities to attend to and appraise their own emotions and the means one chooses to regulate the intensity and duration of emotional states [13,14]. Previous literature shows that effective emotion regulation can help to deal with challenging situations adequately, whereas poor emotion regulation abilities may work as risk factors when facing threatening or uncertain events [15,16]. Since the study of specific emotion regulation strategies does not necessarily account for individual variability, it is recommended to use a broader approach. This goal can be achieved by measuring trait emotional intelligence (trait EI) that targets the flexibility and adaptability of people's regulations, thus leading to a more effective way of assessing emotion regulation as an individual difference [17]. For instance, research on EI is more focused on capturing the outcome of emotion regulation instead of investigating the process behind it, as it could be for studies investigating the different strategies people can use to cope with emotional experiences [18]. Furthermore, people with high (vs. low) EI tend to recognize their emotions at very early stages, they do not have overreactions that require more regulation, and they also manage to adopt emotional regulation strategies flexibly depending on the situation [17]. Previous studies have shown a negative association between emotion regulation and anxiety-related pathologies [16], an effective impact of emotion regulation training to reduce anxiety [19,20] and, also, a positive association between EI and mental health in general [21,22]. With the health emergency caused by the COVID-19 pandemic and increased concerns about people's psychological well-being, there was a need to urgently deal with mental health problems and assess factors that reduce distress and promote resilience. In light of the protective effects of EI in response to stressful events, several studies have been focused on investigating the mental health benefits of this factor during the COVID-19 pandemic [23–25]. However, to the best of our knowledge, there are no studies that have been concerned with investigating how these three factors—state anxiety, trait EI, and risk perception—were correlated and interacted with each other in the first stages of the SARS-CoV-2 pandemic in Italy. Thus, we assessed the impact of risk perception on people's anxiety and the protective role of trait EI at two crucial time points in Italy: immediately after the lockdown was imposed (wave 1) and when the lockdown ended (wave 2), while accounting for the number of new positive cases recorded the day before each participant completed the survey.

## 2. Materials and Methods

### 2.1. Participants Recruitment

For the current study, 2774 participants were recruited in a cross-sectional design, during two temporally different data collections (wave 1, N = 1679; wave 2, N = 1095). The first wave was posted online between March 10 and March 20, 2020, immediately after the Italian government enacted a full countrywide lockdown. The second wave was posted online between 11 May and 31 May 2020, when the lockdown was initially lifted. Both surveys followed the same posting procedures and were disseminated on social media platforms (e.g., Facebook) and through instant messaging (e.g., WhatsApp). Specifically, we created a network of people (e.g., acquaintances, friends, relatives) with the request to share the survey. This network of individuals was diversified by age and gender in order to reach a sample as similar as possible to the general population. All those who helped during the data collection used the same message, previously prepared by the research team. Informed consent was obtained from all participants who were fully informed of the

purpose, risks, and benefits of the study. The research was conducted in accordance with the Helsinki Declaration as revised in 1989.

### 2.2. Measures

After giving their consent to participate in the study, participants were asked to report some demographic data (i.e., age, gender, education level, income, and political orientation). Then, they were asked to fill in two measures of individual differences: the state subscale of the State-Trait Anxiety Inventory (STAI; wave 1, $\alpha$ = 0.92; wave 2, $\alpha$ = 0.93) [26] and the Trait Emotional Intelligence Questionnaire-Short Form (TEIQue-SF; wave 1, $\alpha$ = 0.88; wave 2, $\alpha$ = 0.88) [27]. In particular, the state subscale of the STAI includes 20 items measuring how often participants experienced a range of emotional reactions over a short period of recent time (i.e., within the past two weeks). Participants were asked to answer each item on a 4-point scale ranging from 1 ("almost never") to 4 ("almost always"). The TEIQue-SF includes 30 items measuring people's trait emotional intelligence (trait EI), i.e., their capacity to recognize, express, and regulate their own emotions. Participants were requested to answer each item on a 7-point scale ranging from 1 ("completely disagree") to 7 ("completely agree"). Finally, we measured participants' risk perception of getting COVID-19 with a single item, i.e., "How high do you think the risk of getting sick from COVID-19 is?". Participants were asked to answer on a slider from 0 ("not at all high") to 100 ("very high"). At the end of the survey, participants were also asked to report how religious they were and how much they trusted the authorities. For both these questions, participants were asked to respond on a scale from 1 ("not at all") to 7 ("very much").

Furthermore, the reported number of infected cases per million Italian citizens during the days when the two surveys were posted online was recorded. To achieve this, we retrieved the data from the Our World in Data website [28].

### 2.3. Data Analysis

We ran a multiple regression analysis with a backward model selection procedure. We used an exploratory analysis approach as the current literature lacks knowledge on how the variables of interest impact state anxiety during a pandemic. Specifically, we used the Akaike information criterion (AIC) model comparison to compare sets of candidate models fitted to the same data using the step AIC function in R [29]. The advantages of this approach are the selection of the most plausible model, and the ranking and weighting of the remaining models in a pre-defined set [30]. Results were interpreted in terms of AIC, Akaike weights, significance, size of coefficients, and explained variance. Generally, smaller AIC values are indicators of a more parsimonious model.

## 3. Results

For each wave, following the initial data acquisition, the data from underage participants and those who did not respond from Italy were excluded. Furthermore, only fully completed surveys were included in the analyses. Hence, wave 1 includes 1031 participants (mean age 30.63 years ranging from 18 to 75, 67.99% females), and wave 2 includes 700 participants (mean age 30.97 years ranging from 18 to 75, 69.98% females). Thus, the overall response rate was about 62.80%. For full details on the participants, see Table 1.

Consistent with the pandemic trend and the gradual relaxation of the restrictive measures between the two waves, descriptive statistics showed that risk perception was lower in the second wave than in the first wave. In addition, the correlation between trait EI and anxiety was higher in wave 2 than in wave 1, while the opposite was true for the correlation between risk perception and anxiety (see Table 2).

To assess which factors predicted state anxiety, we ran a backward model selection analysis with a starting model that included the three-way interaction between trait EI, risk perception, and wave, while controlling for age, gender, education level, income, political orientation, religiosity, trust in authorities, and rate of cases per million inhabitants. The final and most plausible model included the main effects of trait EI, wave, risk

perception, age, gender, education level, and political orientation. Further, the interaction between wave and risk perception, and the interaction between trait EI and wave were also significant (see Table 3).

**Table 1.** Characteristics of the participants and responses to the main survey items in the two waves of the study.

| Characteristics | Wave 1 (N = 1031) | Wave 2 (N = 700) | Difference |
|---|---|---|---|
| Age, y (range) | 30.63 (18–75) | 30.97 (18–75) | $t = -0.528$ |
| Gender | | | $X^2 = 2.11$ |
| Female, no. (%) | 701 (67.99%) | 492 (69.98%) | |
| Highest level of education | | | $t = -1.654$ |
| Primary school, no. (%) | 1 (0.09%) | 0 (0%) | |
| Middle school, no. (%) | 50 (4.85%) | 23 (3.27%) | |
| High school, no. (%) | 460 (44.62%) | 306 (43.53%) | |
| Bachelor's degree, no. (%) | 303 (29.39%) | 213 (30.30%) | |
| Master's degree, no. (%) | 187 (18.14%) | 133 (18.92%) | |
| Specialization/Doctorate, no. (%) | 30 (2.91%) | 28 (3.98%) | |
| Income | | | $t = 0.143$ |
| >10,000 (%) | 65 (6.30%) | 37 (5.26%) | |
| 10,000–19,999 (%) | 190 (18.43%) | 124 (17.64%) | |
| 20,000–29,999 (%) | 9 (0.87%) | 7 (1%) | |
| 30,000–39,999 (%) | 260 (25.22%) | 172 (24.47%) | |
| 40,000–49,999 (%) | 153 (14.84%) | 114 (16.22%) | |
| 50,000–59,999 (%) | 93 (9.02%) | 63 (8.96%) | |
| 60,000–69,999 (%) | 50 (4.85%) | 39 (5.55%) | |
| 70,000–79,999 (%) | 29 (2.81%) | 23 (3.27%) | |
| 80,000–89,999 (%) | 27 (2.62%) | 25 (3.56%) | |
| 90,000–99,999 (%) | 17 (1.65%) | 5 (0.71%) | |
| 100,000–109,999 (%) | 3 (0.29%) | 6 (0.85%) | |
| 110,000–119,999 (%) | 7 (0.68%) | 13 (1.85%) | |
| 120,000–129,999 (%) | 3 (0.29%) | 2 (0.28%) | |
| 130,000–139,999 (%) | 3 (0.29%) | 1 (0.14%) | |
| 140,000–149,999 (%) | 0 (0%) | 0 (0%) | |
| >150,000 (%) | 1 (0.10%) | 2 (0.28%) | |
| Prefer not to say (%) | 121 (11.74%) | 70 (9.96%) | |
| Political orientation | | | $t = -1.05$ |
| Extreme left wing, no. (%) | 26 (2.52%) | 19 (2.70%) | |
| Left wing, no. (%) | 285 (27.64%) | 176 (25%) | |
| Center-left wing, no. (%) | 335 (32.50%) | 230 (32.72%) | |
| Center wing, no. (%) | 152 (14.74%) | 103 (14.65%) | |
| Center-left wing, no. (%) | 142 (13.77%) | 107 (15.2%) | |
| left wing, no. (%) | 85 (8.25%) | 58 (8.25%) | |
| Extreme left wing, no. (%) | 6 (0.58%) | 7 (1%) | |
| Religiosity, 1–7 scale (SD) | 2.944 (1.824) | 2.95 (1.791) | $t = -0.152$ |
| Trust authorities, 1–7 scale (SD) | 4.250 (1.357) | 4.124 (1.378) | $t = 4.668$ *** |

Note: *** $p < 0.001$.

Thus, we ran a slope analysis to probe the interaction between risk perception and wave. Findings showed that the relationship between risk perception and anxiety was significant in the first wave (B = 0.095, $p < 0.001$), but not in the second wave (B = 0.019, $p = 0.32$). Thus, in the first wave, the increase in the level of risk perception was associated with an increase in state anxiety (Figure 1). Subsequently, a second slope analysis showed that in both waves, higher trait EI was associate with lower levels of state anxiety. Moreover, this relationship was stronger in the second wave (B = −9.26, $p < 0.001$) compared to the first wave (B = −5.40, $p < 0.001$; Figure 1).

**Table 2.** Descriptive statistics, differences between wave 1 and wave 2, and correlations for wave 1 (below the diagonal) and wave 2 (above the diagonal).

| | | Wave 1 | | | Wave 2 | | | | | |
|---|---|---|---|---|---|---|---|---|---|---|
| | | Mean | SD | Range | Mean | SD | Range | *t* | *d* | 95% C.I. |
| State anxiety | | 46.06 | 13.82 | 20–75 | 45.59 | 14.32 | 20–80 | 0.68 | 0.03 | [−0.06, 0.13] |
| Trait EI | | 4.88 | 0.75 | 2.13–6.83 | 4.75 | 0.76 | 2–6.77 | 3.66 *** | 0.18 | [0.08, 0.28] |
| Risk perception | | 66.33 | 23.66 | 0–100 | 53.11 | 23.88 | 0–100 | 11.37 *** | 0.56 | [0.46, 0.65] |
| Cases per million | | 211.2 | 85.52 | 122–521.1 | 16.79 | 3.43 | 7.88–21.12 | 60.1 *** | 2.94 | [2.81, 3.08] |

Descriptive Statistics

Correlations

| | | Wave 2 | | | |
|---|---|---|---|---|---|
| | | State anxiety | Trait EI | Risk perception | Cases million |
| Wave 1 | State anxiety | — | −0.51 *** | 0.13 *** | −0.10 ** |
| | Trait EI | −0.30 *** | — | −0.15 *** | 0.11 ** |
| | Risk perception | 0.18 *** | −0.04 | — | −0.06 |
| | Cases per million | −0.03 | −0.02 | 0.13 *** | — |

Note: ** $p < 0.01$; *** $p < 0.001$.

**Table 3.** Regression model predicting participants' state anxiety.

| | β | *B* | *SE* | *t* | 95% C.I. |
|---|---|---|---|---|---|
| Trait EI | −0.29 | −5.40 | 0.52 | −10.33 *** | [−6.43, −4.38] |
| Wave | 0.78 | 22.27 | 4.35 | 5.12 *** | [13.74, 30.79] |
| Risk perception | 0.17 | 0.10 | 0.02 | 5.78 *** | [0.06, 0.13] |
| Age | -0.14 | -0.15 | 0.02 | -6.55 *** | [−0.19, −0.11] |
| Gender | −0.12 | −3.38 | 0.61 | −5.58 *** | [−4.57, −2.19] |
| Education | 0.09 | 1.40 | 0.33 | 4.20 *** | [0.75, 2.06] |
| Political orientation | -0.06 | -0.59 | 0.23 | -2.56 * | [−1.03, −0.14] |
| Wave x Risk perception | −0.16 | −0.08 | 0.03 | −2.92 ** | [−0.13, −0.03] |
| Wave x Trait EI | −0.66 | −3.86 | 0.81 | −4.77 *** | [−5.45, −2.27] |
| $AdjR^2 = 0.22$, $p < 0.001$, AIC baseline = 8731.4; AIC model = 8722.52 | | | | | |

Note: * $p < 0.05$; ** $p < 0.01$; *** $p < 0.001$.

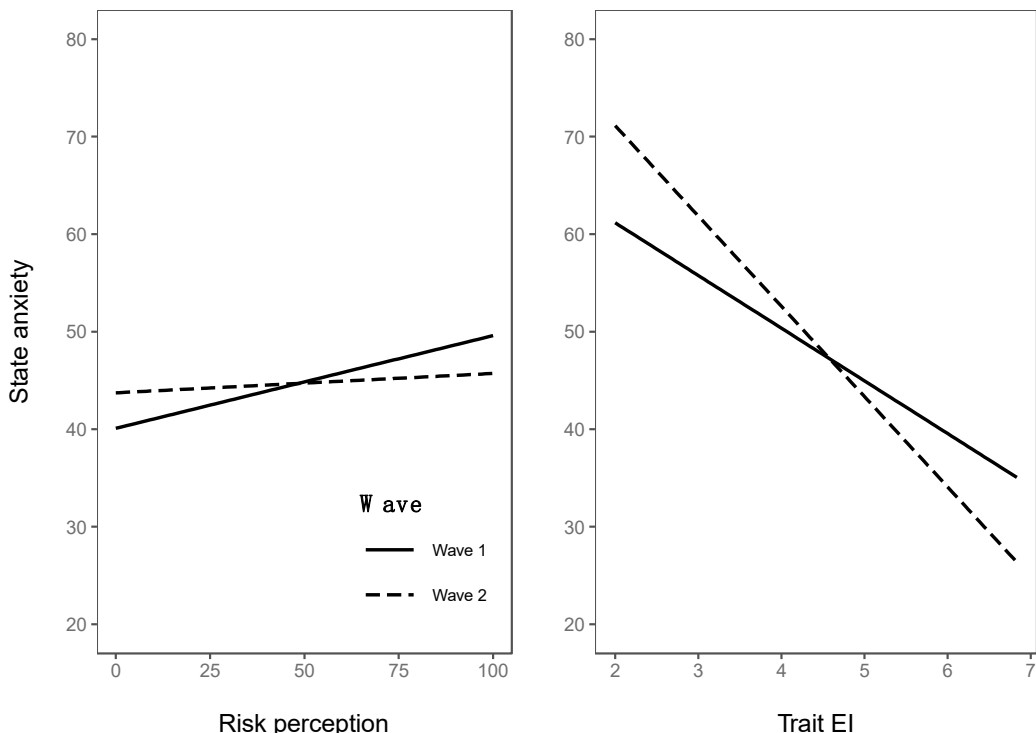

**Figure 1.** Slopes for risk perception (left panel) and trait EI (right panel) in wave 1 and wave 2.

## 4. Discussion

This is the first study to assess the role of trait EI, risk perception, and time on people's state anxiety during the COVID-19 pandemic. In particular, data collections were carried out at two crucial moments of the pandemic outbreak in Italy: when the government imposed drastic restrictive measures in order to contain the contagions by enforcing the full countrywide lockdown (8 March 2020) and when this was withdrawn giving way to a gradual reopening of activities. The results show a positive relationship between COVID-19 risk perception and state anxiety only during the early phase of the pandemic: specifically, people who perceived a higher risk of getting sick from COVID-19, reported higher levels of state anxiety at the same time. This finding is in line with previous literature on risk perception, which points out that an unknown and uncontrollable hazard with potentially catastrophic consequences—such as COVID-19 in the early stages of its outbreak—results in greater public concern [31] and is associated with a negative affective experience [3]. Moreover, this pattern was not replicated at the end of the lockdown. This is probably due to increased perceived control of the pandemic situation inferred from the government's decision to relax the restrictive measures in response to the objective decrease in the number of cases. Furthermore, prior work shows that the number of cases has an impact on people's risk perception [9] and that risk perception predicts anxiety during the early stages of the full lockdown in Italy. It also extends these findings by showing how the relationship between risk perception and anxiety developed once the lockdown ended. For example, people may have experienced anxiety-related symptoms primarily associated with the perceived likelihood of being infected at the beginning of the pandemic, in which the major concern was implementing any efforts to limit contagions, while over the lockdown many other triggers emerged to have an impact in terms of mental health. Indeed, several studies have revealed the existence of multiple predictors of anxiety and general mental health during the COVID-19 pandemic [6,32] and lockdowns specifically [33,34]. On the other hand, a study conducted in Italy at the same time as our work showed a significant positive relationship between state anxiety and risk perception even when restrictive measures were slowly being relaxed [35]. Albeit it is not possible to draw firm conclusions because of the cross-sectional nature of our study, this finding might be interesting to investigate in other global emergencies. For example, while in the short-term anxiety associated with perceived risk might be functional in increasing compliance with protective behaviors [9], in the long-term it might become chronic and have a negative impact on both mental health and perceived effectiveness in coping with the emergency itself [36,37].

Furthermore, regardless of the relationship between risk perception and state anxiety, trait EI was found to work as a protective factor both during the lockdown and at the reopening. In line with the literature, people with higher trait EI had fewer anxiety symptoms [15] and this buffering effect was even stronger during the reopening. The rationale here might be that people with higher trait EI were able to deal better with the pandemic, especially in the long run. That is, their ability to self-regulate allowed them to better maintain their homeostasis and manage emotions effectively, resulting in lower mental health problems especially when they had to adjust to months of lockdown and pandemic. Once again this is in line with previous studies and confirms the importance of developing interventions targeted at promoting emotion regulation strategies when facing not only major sources of stress but also worldwide health emergencies [17,18]. In recent years, the use of EI interventions has increased, and such interventions have been used to improve different outcomes, including life satisfaction, stress reduction, and mental health [38,39]. Besides the personal daily benefits, the application of such interventions in different contexts (e.g., educational, occupational, and health settings) can help in improving society's resilience and capabilities to face even new and global threats, such as a pandemic.

## 5. Conclusions

The present study highlights the important protective role of trait EI when facing different stages of a pandemic, despite the major limitation of being a cross-sectional rather than a prospective design. Moreover, the generalizability of our findings should be considered with caution since our participants cannot be entirely considered a representative sample of the Italian population (e.g., lower average age and non-equal gender distribution) and we did not use any attention checks and future work should assess whether participants were paying attention or not. Despite these limitations, the present work gives an important contribution to the literature by extending knowledge on the moderating role of the individuals' abilities to recognize and manage their own emotions (i.e., trait EI) on the development of anxiety problems in a completely unprecedented situation as the current COVID-19 pandemic. It also gives some important advice for policymakers and mental health professionals suggesting the promotion of prevention protocols among the general population aimed at increasing these abilities. For instance, a way to achieve this goal would be by developing an application that monitors people's physiological and psychological reactions to a stressful situation and provides them with basic advice on how to break a cycle that can lead to increased anxiety over time.

Still, although no conclusive inferences can be drawn about the relationship between state anxiety and risk perception, this finding deserves further investigation within the risk perception literature. Indeed, while two main modes in which people may interact with a hazard are defined within this framework—risk as feelings and risk as analysis [40]—, the affective component appears to be a stronger predictor of behavior than the cognitive one (for example, [41]). However, the long-term effects of negative emotions in response to even other global emergencies are currently being debated [36,37]. Once again, interventions aimed at enhancing emotion regulation skills could be crucial in achieving such goals.

**Author Contributions:** Conceptualization, E.T. and N.O.; methodology, E.T. and N.O.; formal analysis, E.R. and S.S.; investigation, E.T., N.O. and A.H.; data curation, E.T., N.O. and A.H.; writing—original draft preparation, E.T., N.O., A.H., E.R. and S.S.; writing—review and editing, E.T., N.O., A.H., E.R. and S.S.; supervision, E.R. and S.S. All authors have read and agreed to the published version of the manuscript.

**Funding:** This study was supported in part by the CARIPARO (Cassa di Risparmio di Padova e Rovigo) Foundation research fellowship.

**Institutional Review Board Statement:** The study was conducted in accordance with the Declaration of Helsinki.

**Informed Consent Statement:** Informed consent was obtained from all subjects involved in the study.

**Data Availability Statement:** The data that support the findings of this study are available from the corresponding author, S.S., upon reasonable request.

**Conflicts of Interest:** The authors declare no conflict of interest.

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
