# Peer review of "Trends in State Anxiety during the Full Lockdown in Italy: The Role Played by COVID-19 Risk Perception and Trait Emotional Intelligence"

_traumacare, doi:10.3390/traumacare2030034_

Round 1

Reviewer 1 Report

Generally speaking, this article is a meaningful paper.

1. This paper involves several key concepts, such as anxiety and risk perception, but in the introduction part, I don't think the author has made a  comprehensive and organized review.

2. In the materials and methods section, the problems are as follows:

(1) This paper conducted two rounds of investigation, but the author did not clearly express in the text whether the investigation passed the ethical review in advance or not. This is very important and necessary for a  psychological investigation. I think it is a  neglect of the author.

(2)The lack of necessary methodological description makes me feel that the research design of this paper is not clear. What are the research steps to follow?

(3)How is risk perception quantified?

(4)How to control the quality of survey data is also not explained in the article.

Author Response

  1. We understand what you mean and are sorry the introduction fell short of your expectations. We amended the paper to more comprehensively cover the key concepts and we hope you will find the introduction improved. 
    1. We agree with you that this is a critical issue. Yet this study was pre-prepared and we aimed to assess individuals’ responses to the spread of the virus immediately after it became a threat to the Italian population. Data collection began a few days after the government forced the lockdown. All was very sudden and we did not have the time to ask the ethical committee for their approval because that would have required too much time and we thought that collecting data from the very beginning of the pandemic was important. We understand that this is a methodological flaw, however as for interventions during emergencies also research must have the flexibility to respond quickly in order to gather important information.  Hence, we exceptionally did not request the University's ethical approval. We did, however, strictly apply all the rules requested by the ethical committee and used the same rigor in respecting ethical norms.
    2. Thanks for pointing this out. We agree with you that the methods lacked some important information and it was hard to follow for the readers. We extended the materials and methods section in order to give more details and information about the design and the procedure. We hope that what we have done is now clearer.
    3. You are right and we should have been more explicit about this important measure. On lines [109-111], we now report the question measuring risk perception verbatim and the response scale the participants used to answer it.
    4. We did not use any attention checks. However, we think that the lack of external incentives to participate and the indirect method of recruitment (partially unrelated to experimenters) may sufficiently reflect the quality of the data. And we excluded from the analyses all those who didn't fill out the survey until the end.

Reviewer 2 Report

Thank you for the opportunity to review this article. The authors present a short, clear research study with a good-sized sample from a population which was highly impacted by COVID-19 in the early phase of the pandemic.

I have a few comments:

  • On Lines 158-166, the authors hypothesize a reason for the absence of a relationship between risk perception and anxiety at Wave 2, i.e., that people "were aware that conditions had improved". This interpretation goes beyond the data, and the authors should be more cautious in suggesting that this is the reason. They cannot know that this is the case.
  • Line 138: you can't say that EI (or any other variable) had an "effect" or an "impact" on anxiety. You measured relationships in a correlational design, not an experimental design, so you can only say that variables are related, not that there is a causal effect of one on another.
  • I was confused about whether EI was the same as ER or whether these were two different variables, as on Line 128, the authors state that the interaction between EI and wave was significant (Table 1), but the table shows emotion regulation, not EI. Please clarify

Author Response

  • [Old lines 158-166, current lines 174-183] We would like to thank you for pointing this out. Of course, we are offering a possible interpretation but, as you wrote, it goes beyond what our data can say. We stand by our interpretation but have amended the paper to make clear that the interpretation is speculative and future work should investigate it and provide the evidence that we do not have available at this time.
  • [Old line 138, current lines 154-155] You are right, we should have been more careful. We edited the sentence in order to avoid inferring a causal effect between variables. 
  • Thanks for highlighting this inconsistency in our labels. We fixed the table accordingly by using the label Trait EI (i.e., Trait Emotional Intelligence).

Round 2

Reviewer 1 Report

On the basis of the first version, the author has made a good improvement on my previous opinions, especially in the methodology part.  But I think there are still some details that have not been explained very clearly.

  1. First of all, I think some improvements should be made in the introduction. I don't think the author has arranged this part logically. The author is elaborating according to his own ideas, but it makes me feel that I still can't well understand the two core concepts of risk perception and emotion regulation.
  2. The author made some supplementary explanations for the methodology part. I can probably understand the whole experimental process.
  3. In the results section, I don't understand why the author lacks statistics of key information such as demography? And there seems to be no attempt to explore the relationship between demographic information and the core concept home.
  4. In the discussion part, I still think some words are not refined enough. I think the author should make better improvements in his writing techniques in order to make this part of the text easier to read,for example, from line 161 to line 167.
  5. In the conclusion, I think the author can further explain the research significance of this paper, especially some practical significance.

Author Response

  1. We would like to thank the reviewer for raising this point. We agree that focusing more on the two key constructs of our study and making this part easier to follow for the reader is very important. Therefore, in the introduction, we have now better clarified the concepts of risk perception and trait emotional intelligence. Moreover, we amended the paper to streamline the introduction and make it more logically consistent.
  2. We provided some more information about the measures we used and reported them in the tables. We hope that our findings are now easier to follow and understand for the readers. 
  3. Thanks for pointing this out. We agree with the reviewer and moved the table in the main text. Further, we now report in the paper the descriptive statistics related to the demographic information. 
  4. We would like to thank the reviewer for this comment. We modified the discussion section substantially by dividing the explanation of the main results into two sections: in the first part, we focused on discussing the relationship between state anxiety and risk perception in the first wave and providing some possible interpretations as to why this might not have occurred in the second wave; in the second part, instead, we discussed the moderation of trait emotional intelligence on state anxiety.
  5. We agree with the reviewer that it is important to explicitly present the possible implications of the study and thus we added some information regarding possible emotional intelligence interventions, the benefits they can provide, and how they are connected to global threats such as the pandemic.

Round 3

Reviewer 1 Report

1. The article results cannot interpret the title and the result does not make it clear that people " know how to regulate emotions ".

2. The results of Table 2 and slope analysis are partially repeated.

3. The results show that wave, EI traits, risk perception, gender, education level, and income are all included in the model, and the population part is also described. Why are the gender, education level, and income in some models directly thrown away? Can we do further analysis? (just like analyzing wave)

Author Response

  1. We thank the reviewer for raising this point. We have changed the title as follows: “Trends in state anxiety during the full lockdown in Italy: The role played by COVID-19 risk perception and trait emotional intelligence” 
  2. Table 2 shows the correlations between the variables, while the slope analyses, in the text, probe the interactions that were found to be significant from the backward model selection analysis (and we reported the coefficient ‘b’, not the Pearson's ‘r’). Both information are necessary to report the data in a correct way. However, we acknowledge that describing the results of slope analysis in terms of correlations could be confusing. We hope the Reviewer agrees with these changes.
  3. To assess which factors predicted state anxiety, we ran a backward model selection analysis with a full starting model that included all predictors: the three-way interaction between trait EI, risk perception, and wave, while controlling for age, gender, income, education level, political orientation, religiosity, trust in authorities, and rate of cases per million inhabitants. As it is done in this type of analysis, the best model was selected based on the Akaike Information Criterion (AIC). This is an explorative type of analysis, which was well suited to the new and unfamiliar conditions created by the coronavirus pandemic. Therefore, some variables simply did not fit in the final and most plausible model resulting from this analysis (i.e. political orientation, religiosity, trust in authorities, age, and rate of cases per million inhabitant), based on the AIC (starting AIC = 7587.04, final model AIC = 7582.13). This is the reason why they were discarded from the final model and it did not depend on a decision made by us.